# Safety and Immunogenicity of a *Shigella* Bivalent Conjugate Vaccine (ZF0901) in 3-Month- to 5-Year-Old Children in China

**DOI:** 10.3390/vaccines10010033

**Published:** 2021-12-28

**Authors:** Yi Mo, Wenjian Fang, Hong Li, Junji Chen, Xiaohua Hu, Bin Wang, Zhengli Feng, Honghua Shi, Ying He, Dong Huang, Zhaojun Mo, Qiang Ye, Lin Du

**Affiliations:** 1Guangxi Center for Disease Prevention and Control, Nanning 530027, China; moyi0415@hotmail.com (Y.M.); chenjunji2013@163.com (J.C.); fengfir@163.com (Z.F.); mozhj@126.com (Z.M.); 2Beijing Zhifei Lvzhu Biopharmaceutical Co., Ltd., Beijing 100176, China; fangwenjian@zhifeishengwu.com (W.F.); huxiaohua@zhifeishengwu.com (X.H.); 3National Institutes for Food and Drug Control, Beijing 102629, China; lihong@nifdc.org.cn (H.L.); binwang@nifdc.org.cn (B.W.); qiangyee@nifdc.org.cn (Q.Y.); 4Yongfu Center for Disease Prevention and Control, Guilin 541800, China; yfshh2006@126.com (H.S.); heying0217@126.com (Y.H.); 5Yizhou Center for Disease Prevention and Control, Hechi 546399, China; 15878860518@163.com; 6Beijing Bacterial Vaccine Engineering Research Centre, Beijing 100176, China

**Keywords:** *Shigella* conjugate vaccine, bivalent, safety and immunogenicity, clinical trial, infants and young children

## Abstract

No licensed *Shigella* vaccine is presently available globally. A double-blinded, randomized, placebo-controlled, age descending phase II clinical trial of a bivalent conjugate vaccine was studied in China. The vaccine ZF0901 consisted of O-specific polysaccharides purified and detoxified from lipopolysaccharide (LPS) of *S. flexneri* 2a and *S**. s**onnei* and covalently bonded to tetanus toxoid. A total of 224, 310, and 434 children, consented by parents or guardians, aged 3 to 6 and 6 to 12 months and 1 to 5 years old, respectively, were injected with half or full doses, with or without adjuvant or control Hib vaccine. There were no serious adverse reactions in all recipients of ZF0901 vaccine independent of age, dosage, number of injections, or the adjuvant status. Thirty days after the last injection, ZF0901 induced robust immune responses with significantly higher levels of type-specific serum antibodies (geometric mean concentrations (GMCs) of IgG anti-LPS) against both serotypes in all age groups compared with the pre-immune or the Hib control (*p* < 0.0001). Here, we demonstrated that ZF0901 bivalent *Shigella* conjugate vaccine is safe and immunogenic in infants and young children and is likely suitable for routine immunization.

## 1. Introduction

Shigellosis remains a common, serious, and increasingly difficult-to-treat disease, especially because of the emergence of antibiotic-resistant strains [1,2,3,4]. For example, a survey in 2016 showed that there were an estimated more than 269 million cases of shigellosis with over 212,000 deaths, of which about 74 million cases of shigellosis were in children younger than 5 years old and caused more than 63,000 deaths [5]. In mainland China, the annual morbidity of bacillary dysentery ranked as one of the top five notifiable infectious diseases, according to the Chinese Center for Disease Control and Prevention (CDC) [6,7,8]. Although the incidence rate of bacillary dysentery has declined markedly due to the improvement of sanitation and hygiene, there were over 80,000 dysentery cases in China in 2020 [9]. Similar to the global trend, the highest incidence of Shigellosis in China is among children under 5 years old and the prevalent serotypes are *S. flexneri* and *S. sonnei*. Vaccine research against *Shigella* has been under development for several decades; unfortunately, there is, so far, no licensed vaccine available [10,11,12,13].

*Shigella* is a rod-shaped, Gram-negative pathogen. There are four species of *Shigella* commonly isolated from patients with shigellosis; among them, *S. flexneri* and *S. sonnei* are predominant [14,15,16]. In Africa and Asia, *S. flexneri* and *S. sonnei* are responsible for approximately 90% of shigellosis cases [17]. *S. flexneri* 2a is the dominant serotype among *S. flexneri* and, from early clinical studies, antibodies induced by type 2a conjugate vaccine are cross-reactive with other *S flexneri* serotypes including *S. flexneri* 6 [17,18]. We reason that a novel bivalent vaccine containing both *S. flexneri* 2a and *S. sonnei* can offer broad coverage against the most common pathogenic S*higella* serotypes.

The LPS of *Shigella* spp. is a virulence factor and its O-polysaccharide component (O-antigen) serves as one of the promising protective antigens. Several *Shigella* conjugate vaccines based on O-antigen have been evaluated in clinical trials [13]. For instance, an investigational vaccine against *S. sonnei* showed 74% protection in young adults [19]. Similarly, the vaccines against the *S. flexneri* 2a serotype showed safety and immunogenicity in different age groups [20,21]. Up until the present, most *Shigella* conjugate vaccine candidates consist of a single serotype.

There are advantages of using O-antigens extracted from bacteria. The purification involves minimum physical-chemical processing and is likely to retain the native carbohydrate structures, in particular, the non-reducing end sugars. In Gram-negative organisms, the immunodominant epitopes are located at the non-reducing end. This fact is best demonstrated in *Vibrio Cholerae,* where the methylation of the non-reducing end is the serotyping marker that distinguishes the Ogawa from Inaba serotypes [22].

Previously, we conducted a preliminary Phase I study of a bivalent *S. flexneri* 2a and *S. sonnei* vaccine (ZF0901), consisting of O-antigens covalently bound to tetanus toxoid (TT), in an age-descending order from adults to infants 3 months old (Clinicaltrials.gov identifier: NCT03561181) (unpublished). This safety screening was a prerequisite for subsequent clinical trials following the guidelines of the Chinese National Medical Product Administration (NMPA). The results demonstrated that ZF0901 was safe in all studied age groups, and permission for further investigation was subsequently granted.

Here we present a phase II safety and immunogenicity study of a bivalent *Shigella* conjugate vaccine ZF0910 in children 3 months to 5 years old. The vaccine was administered more than 7 days apart from any routine vaccination in infants and toddlers. The effects of adjuvant and dosage (full or half dose) were also investigated.

## 2. Materials and Methods

The Investigational New Drug and One-time Certification for Clinical Investigation were approved by NMPA (No. 2015L00057 and No. 2018Y36); the study protocol and informed consent were approved by the Medical Ethics Committee of Guangxi (No. GXIRB2018-0017). Written, informed consent was obtained from parents or guardians of trial candidates before screening. The trial was registered at Clinicaltrials.gov (Identifier: NCT04865497). This study was undertaken in accordance with the standards of the International Conference on Harmonization guidelines on Good Clinical Practices (ICHGCP), and followed the ethical principles established in the Declaration of Helsinki.

### 2.1. Study Design and Clinical Protocol

The present ZF0901 phase II clinical trial was a double-blinded, randomized, and placebo-controlled study in children aged 3 months to 5 years old in a single geographic region. The randomization was generated with SAS (SAS 9.4) on blind bases. The site of the study was at Yongfu County, Guangxi Zhuang Autonomous Region in the Southwestern part of China, approximately 330 Km from the regional capital of Nanning. Children in the area are prone to diarrhea diseases, including shigellosis, due to the region’s subtropical climate and frequent flooding. It is an agriculture-based community, with a population of 280,000 and birthrate of approximately 1.3% per year.

All potential participants must have met all of the following inclusion criteria to be eligible for enrollment into the study: healthy children 3 months to 5 years of age whose parents or legal guardians were willing and competent to provide signed and dated written, informed consent after being informed of all pertinent aspects of the study. Key exclusion criteria were children with known allergic reaction to tetanus toxoid or who had a history of severe adverse reaction associated with a vaccine and/or severe allergic reaction (e.g., anaphylaxis) to any component of the study vaccine. Children participating in any other studies during the investigation period was also excluded. Children with immunodeficiency diseases or who administered immunosuppressive agents, with serious chronic disorder, had history of craniocerebral trauma, encephalopathy, and psychosis, or any other disorder that, in the investigators’ opinions were not compatible with the study, were excluded from participation.

Criteria for temporary delay of vaccine administration included febrile illness (axillary temperature ≥38 °C within 3 days) and other acute illness within 48 h before administration of investigation vaccines. Children with diarrhea or bloody stool within 3 days were also temporarily disqualified.

Children recruited were divided into three age groups: 3 to 6 months, 6 to 12 months, and 1 to 5 years old and further divided randomly into 12 subgroups for different vaccine dosages, formulation, or number of injections. In cohort 1, 250 young infants, aged 3 to 6 months old, were randomly assigned in a 2:2:1 ratio to receive a half or full dose of the trial vaccine or *Haemophillus*-TT conjugate (Hib vaccine) as placebo. Each child received three intramuscular (IM) injections on days 0, 30, and 60. In cohort 2, 350 infants, aged 6 to 12 months old, were randomly assigned in a 2:2:2:1 ratio to receive a half or full dose vaccine, full dose vaccine without adjuvant, or Hib vaccine as placebo. Each child received two injections on days 0 and 30. In cohort 3, 350 toddlers, aged 1 to 5 years old, were randomly assigned in a 2:2:2:1 ratio to receive one injection of a half or full dose, full dose vaccine without adjuvant, or Hib vaccine as placebo. In order to evaluate the effect of additional injection in toddlers, a separate cohort with 100 children, 1 to 5 years old, was also recruited to receive two injections of a full dose on days 0 and 30.

Administration of vaccines and collection of blood samples were performed at the investigation site by appropriately qualified, Good Clinical Practice (GCP)-trained, and vaccine-experienced designated staff members (e.g., physician, nurse, physician’s assistant, nurse practitioner, pharmacist, or medical assistant) as allowed by local, state, and institutional guidelines. Details of vaccine administration of each vaccinee were recorded on a Case Report Form (CRF).

On the day of inoculation, children were examined by the health staff and their axillary temperatures were measured. Those without signs of infection and who had a normal temperature received a single dose (0.5 mL) of the study vaccine, administered intramuscularly into the deltoid muscle. The vaccinees were under observation on site for at least 30 min after injection and body temperatures were taken prior to departure.

Blood samples were collected before the first and 30 days after the last injection via venipuncture.

### 2.2. Vaccine

The investigational vaccine ZF0901 was developed by Beijing Zhifei Lvzhu Biopharmaceutical Co., Ltd. (Beijing, China) and manufactured in accordance with the current Good Manufacture Practices. ZF0901 consisted of the O-SP, purified from LPS of *S. flexneri* 2a or *S. sonnei*, which was covalently bound to TT through a linker adipic acid dihydrazide to form conjugates. The residual endotoxin content in the final product was lower than 10 EU/µg polysaccharide, which met the requirements of the product approved by NMPA. The injection volume was 0.5 mL and each full dose contained 10 µg of *S. Sonnei* O-SP, 10 µg of *S. Flexneri* 2a O-SP, and 50 µg of TT. Those adsorbed on aluminum phosphate as adjuvant contained 0.4 mg Al^3+^/mL. For the half-dose study, each vial contained 5 µg of *S. Sonnei* O-SP, 5 µg of *S. Flexneri* 2a O-SP, and 25 µg of TT.

The control vaccine injected into the placebo group, Hib-TT, was a licensed commercial product (Beijing Zhifei Lvzhu Biopharmaceutical; lot 201806001) containing 10 µg of Hib capsular polysaccharide (CP) conjugated to 24 µg of TT. Both the investigational vaccine and the placebo were packed in single-dose, prefilled syringes, labeled according to the study requirement with individual tracking bar codes for blinding. The vaccines were stored at 2–8 °C with continuous-monitoring temperature systems.

### 2.3. Randomization and Blinding

Statisticians from a certified, independent company used SAS software (SAS 9.4) to generate a randomized, blinding base for the three age groups. Among them, 3- to 6- month-old subjects were randomly assigned to three groups in the ratio of 2:2:1, 6- to 12- month-old subjects were randomly assigned to four groups in the ratio of 2:2:2:1, and 1- to 5- year-old subjects were assigned to five groups in the ratio of 2:2:2:2:1. Investigators at the trial site assigned study numbers to each age group according to the screening sequence of the eligible subjects.

Parents, guardians, participants, study coordinators, study-related personnel, trial investigators, and sponsors were masked to the trial-group assignments. The statisticians responsible for randomization were not allowed to participate in other related work of these clinical trials or to disclose the blinding code to any personnel participating in this trial. Each prefilled syringe contained a unique code that ensured appropriate masking.

After completion of the serological analysis, the unblinding was conducted by the participating primary investigators in the presence of National Institutes for Food and Drug Control and Center of Disease Control, Guangxi, personnel.

### 2.4. Outcomes

#### 2.4.1. Assessment of Safety

We assessed the safety endpoints in the study population, including all participants who received at least one dose of vaccination.

Participants were observed on site for at least 30 min after each injection. Guardians of participants, using a paper-based memory aid, kept daily records of local and systemic adverse events for 30 days following each injection and up to 6 months for any serious adverse events. Any serious adverse reaction occurring during the 6 months after the last injection should have been reported to the local drug administration, the ethics committee, the sponsor, and the clinical trial responsible parties. Laboratory personnel directly conducting the safety compilation remained blinded until the last subject completed the 6-month visit.

#### 2.4.2. Assessment of Immunogenicity

We assessed immunogenic endpoints in the per-protocol population, which included all participants who completed their assigned vaccination schedules and with available antibody results.

Blood samples were collected before the first injection (day 0) and 30 days post the last vaccination. Sera were separated immediately and stored at −20 °C until assayed. The specific concentrations of IgG antibody against *S. sonnei* and *S. flexneri* 2a LPS were measured by ELISA at the National Institutes for Food and Drug Control, Beijing, China. Serum reference was pooled sera from adult donors with high levels of anti-LPS IgG and was assigned to contain 100 ELISA units (EU) anti-LPS IgG per milliliter. Antibody concentrations were expressed in EU compared with the serum reference. We calculated GMCs and corresponding 95% CIs on the basis of standard normal distribution of the antibody concentrations. The minimum detectable level was 0.01 EU for *S. flexner**i* 2a and 0.01 EU for *S. sonnei*. If the comparison among groups showed significant difference, we then performed pairwise comparisons. Hypothesis testing was two sided and we considered *p* values of less than 0.05 to be significant. The seroconversion rate was defined as greater than 4-fold rises. Laboratory personnel directly conducting the immunogenicity assays remained blinded throughout sample assay and data analysis. Unblinded immunogenicity data were analyzed at the completion of the last blood sample. The analyses were descriptive and informed to all participating parties for internal program development decisions and potential support regulatory interactions.

#### 2.4.3. Statistical Analysis

The immunogenicity against *S. sonnei* and *S. flexneri* 2a was evaluated, respectively, according to the anti-LPS IgG levels. In each study group, the serological conversion rate (≥4 fold increase) was calculated individually. The two-sided 95% confidence interval was calculated by using the Clopper–Pearson method. The chi-square test or Fisher’s exact probability tests was used to test differences between the groups. If there were statistical differences between the groups, further statistical analysis would be done. Geometric mean and two-sided 95% confidence interval were used to describe the GMC of antigen-specific IgG antibody after immunization and its growth was multiply compared with that before immunization. Analysis of variance after logarithmic transformation was used to test the difference between the two groups.

Medical Dictionary for Regulatory Activities (MedDRA) as a multi-axial, five-tiered hierarchical terminology was used for medical coding of adverse events and serious adverse events. The number and incidence of all adverse events and adverse events related or not related to the vaccine in each group were calculated, respectively. Fisher exact probability test was used to compare the differences between the groups. The severity, time distribution, and dose distribution of adverse events and the relationship with the study vaccine were described statistically. The adverse events after each dose were statistically analyzed.

## 3. Results

### 3.1. Study Participants

A total of 1204 children were screened in this study, of which 154 were excluded for failing the criteria, 1050 were enrolled, and 959 subjects completed the trial. The diagram depicts screening, enrollment, allocation of vaccine, and dose completion (Figure 1).

Demographic data of study participants by age groups are given in Table 1. In the 3- to 6-months-old group, the mean age of infants was 4.12 months (range 3.0 to 5.9 months), there was a total of 141 (56.4%) male and 109 (43.6%) female. There were no significant differences in mean age (*p* = 0.61) or gender (*p* = 0.67) among each sub-group. In the 6- to 12-months-old group, the mean age of children was 8.36 months (range 6.0 to 11.9 months), 177 (50.7%) male and 172 (49.3%) female. There were no significant differences in gender (*p* = 0.35) but there were differences in mean age (*p* = 0.0005) among each sub-group. In the 1- to 5-years-old group, the mean age of children was 2.49 years (range, 1 to 5.9), with 204 (45.7%) male and 242 (54.2%) female. There were no significant differences in mean age (*p* = 0.66) or gender (*p* = 0.22) among the different vaccine cohorts.

### 3.2. Adverse Reactions

No serious adverse reactions related to vaccines were noted. Fever accounted for the majority of adverse reactions (Figure 2A).

For children aged 3 to 6 months old, a total of 123 (49.2%) of 250 recipients had at least one adverse reaction in all injections, among them 48 (48%) in the half-dose group, 50 (50%) in the full-dose group, and 25 (50%) in the Hib group. There was no statistical difference in the incidence rate between groups by the Fisher exact test (*p* = 0.96).

In the children aged 6–12 months old, 171 (49.0%) of 349 recipients had at least one adverse reaction of all injections, among them 56 (55.5%) in the half-dose group, 47 (47.0%) in the full-dose group, 42 (42.4%) in the full-dose without adjuvant group, and 26 (52.0%) in the Hib group. There was no statistical difference in the incidence rate between groups by the Fisher exact test (*p* = 0.29).

In the children aged 1 to 5 years old, 118 (26.5%) of 446 recipients had at least one adverse reaction of all injections, among them 29 (29.0%) in the half-dose group, 19 (19.4%) in the full-dose, 23 (23.2%) in the full-dose without adjuvant, 17 (34.0%) in the Hib group, and 30 (30.3%) in the full-dose 2 injection group. There was no statistical difference in the incidence rate between groups by the Fisher exact test (*p* = 0.22).

The adverse reactions were also followed and compared after each injection (Figure 2B). Similar adverse reaction rates were observed at each injection in all three age groups; the majority of adverse reactions were fever.

### 3.3. Immunogenicity

There were detectable pre-vaccination antibodies to both *S. flexneri* 2a and *S. sonnei* LPS in all the age groups (Figure 3). The antibody titers decreased as the age increased and the differences between age groups were statistically significant (age 3 to 6 months vs. 6 to 12 months or 1 to 5 years; *p* < 0.001; 6 to 12 months vs. 1 to 5 years; *p* < 0.002). The pre-existing LPS antibodies declined faster with age for *S. sonnei* (all *p* < 0.0001).

#### 3.3.1. 3 to 6 Months Old

Thirty days after the third injection, more than half of the infants who received ZF0901 showed a 4-fold seroconversion in both serotypes (57.7% for *S. flexnari* 2a and 60.7% for *S. sonnei*), and the conversion rates were significantly higher than the Hib control group (*p* < 0.0001). There was no difference in the seroconversion rate between groups receiving full or half doses of ZF0901 (*p* = 0.89 for *S. flexneri* 2a; *p* = 0.07 for *S.*
*sonnei*) (Table 2).

GMC of serum anti-LPS IgG after three injections showed ZG0901 induced a significantly higher response to both serotypes compared with their respective pre-immune levels (for *S. flexnari* 2a half dose, 3.68 vs. 0.45, or full dose 2.85 vs. 0.48, *p* < 0.0001; for *S. sonnei* half dose, 5.67 vs. 0.43, or full dose 3.23 vs. 0.60, *p* < 0.0001) for the group receiving Hib (*p* < 0.0001). When comparing the dosage effect, there was no statistically significant difference between groups that received a full or half dose for *S. flexneri* 2a (3.68 vs. 2.85, *p* = 0.24). However, for *S. sonnei*, the group that received a half dose responded with higher GMC antibodies than those that received a full dose (5.67 vs. 3.23, *p* = 0.01) (Table 2).

#### 3.3.2. 6 to 12 Months Old

More than half of the children in this age group who received ZF0901 responded with a 4-fold seroconversion to *S. flexneri* 2a (64.9%) and *S. sonnei* (86.4%) after two injections, and the seroconversion rates were significantly higher than those in the Hib control (*p* < 0.0001). The comparison between groups vaccinated with ZF0901 (half or full dose, or vaccine without adjuvant) showed no statistically significant difference in seroconversion rates across groups (all *p* > 0.4) (Table 3).

The levels of serum GMC anti-LPS IgG in children injected with ZF0901 were significantly higher than those in pre-vaccination or the Hib group (*p* < 0.001). There were no significant differences in GMC between the groups receiving a full or half dose, with the half dose group responding slightly higher for *S. flexneri* 2a, half vs. full, 4.26 vs. 3.08, *p* = 0.17 and for *S. sonnei*, 6.85 vs. 5.63, *p* = 0.32. When comparing the antibody response in the group injected with ZF0901 containing no adjuvant, it was noticeable that this group had lower antibody levels than groups with adjuvant. Comparing the adjuvant effect in serotype *S. flexneri* 2a, the results were half and full without adjuvant, 4.26 vs. 2.59, *p* = 0.06 and full and full without adjuvant, 3.08 vs. 2.59, *p* = 0.52. A similar trend was found for *S. sonnei* (half and full without adjuvant, 6.85 vs. 6.01, *p* = 0.49; full and full without adjuvant 5.63 vs. 6.01, *p* = 0.75) (Table 3).

#### 3.3.3. 1 to 5 Years Old

Most of the ZF0901 recipients showed 4-fold seroconversion after vaccination (89.4% for *S. flexneri* 2a, 95.5% for *S. sonnei)* and the conversion rates were significantly higher than for the control Hib group (*p* < 0.0001). The rates were similar among vaccine groups: full or half dosages, with or without adjuvant (*S. flexneri* 2a: half and full *p* = 0.09, half and full without adjuvant *p* = 0.96, full and full without adjuvant *p* = 0.10; *S. sonnei*: half and full, *p* = 1.00, half and full without adjuvant, *p* = 1.00, full and full without adjuvant, *p* = 1.00) (Table 4).

For both *S. flexneri* 2a and *S. sonnei*, the GMC anti-LPS IgG levels in all ZF0901 groups were significantly higher than those in pre-vaccination or the Hib group (*p* < 0.0001). There was a dosage effect in antibody response for *S. flexneri* 2a: The group receiving a full dose elicited significantly higher antibody levels than those that received a half dose (8.06 vs. 4.64, *p* = 0.03). In contrast, the dosage effect was not obvious for *S. sonnei* (full vs. half, 9.06 vs. 7.59, *p* = 0.47). The effect of alum adjuvant was also different between the two serotypes. The *S. flexneri* 2a antibody concentration was much higher in the group injected with the vaccine containing adjuvant compared with the group without (8.06 vs. 5.10, *p* = 0.04); however, the effect was reversed for *S. sonnei* (with adjuvant vs. without, 9.06 vs. 16.64, *p* = 0.02) (Table 4).

We included a separate cohort to evaluate the effect of an additional injection given 30 days after the first. The GMC anti-LPS IgG levels were significantly higher in recipients after the second injection (for *S. flexneri* 2a, 17.03 vs. 8.06, *p* = 0.0006; *S. Sonnei*, 13.5 vs. 9.06, *p* = 0.10), but the 4-fold conversion rates compared with those receiving a single injection were not statistically significant (for *S. flexneri* 2a, *p* = 0.46; *S. Sonnei*, *p* = 0.72).

#### 3.3.4. Antibody Response in All Three Age Groups

We compared the immune response across all three age groups following the designated immunization dosage and schedule (Figure 4). The GMC anti-LPS IgG levels for both *S. flexneri* 2a and *S. sonnei* showed an age-dependent response. The specific IgG response was higher in children 1 to 5 years compared with younger age groups, especially in children immunized with a full dose. In groups receiving a full dose with adjuvant, statistical differences could be seen between the 1 to 5 years group and the corresponding younger groups (for *S. flexneri* 2a, 3–6 mo vs. 1–5 yr, *p* < 0.0001, 6–12 mo vs. 1–5 yr, *p* = 0.03; *S. Sonnei*, 3–6 mo vs. 1–5 yr, *p* = 0.001, 6–12 mo vs. 1–5 yr, *p* = 0.005). Comparing groups that received a full dose without adjuvant, the GMC anti-LPS IgG levels were also higher in 1- to 5-year-old children than younger groups, but they were only significant for *S. Sonnei* (*p* < 0.0001).

## 4. Discussion

There appears to be a strong correlation between the level of serum lipopolysaccharide (LPS)-specific IgG antibodies and serotype-specific immunity against *Shigella* [12]. In the current pipeline of vaccine development, a number of O-antigen-based conjugate vaccines were in clinical trials and showed promising immunity against the disease, including bioconjugate vaccine [23,24], synthetic vaccine [25], and detoxified LPS conjugate vaccines [26,27,28]. We designed a bivalent conjugate vaccine, ZF0901, to elicit high levels of long-lived serum IgG O-SP antibodies to both *S. flexenri* 2a and *S. sonnei* formulated to be suitable for routine immunization.

This study showed that conjugate vaccine ZF0901 was safe and immunogenic against both serotypes in neonatal and young children. There were no serious adverse reactions in all recipients of ZF0901 independent of age, number of injections, or the adjuvant status. Fever was the most common side reaction in all age groups: overall, 35.8% vs. 41.3% (*p* > 0.2) in ZF0901 participants and Hib controls, respectively. The rate of fever was similar in all groups independent of dosage, the adjuvant status, or the number of injections. In all three age groups the overall incidence rate of diarrhea, nausea/vomiting, and dysphoria/irritability were 3.7%, 1.9%, and 1.8% in vaccine groups and were not significantly different from the Hib control. The unsolicited ADRs with incidence rates of 0.4% in ZF0901 vaccinees were similar to that of Hib control.

The ZF0901 vaccine elicited a statistically significant rise of type-specific IgG antibodies against *S. flexneri* 2a and *S. sonnei* 30 days post-immunization in all vaccine groups. More than 50% of the recipients had >4-fold seroconversion in all age groups regardless of the dosage or number of injections. There was no significant dosage effect observed. There seemed to be an indication that the effect of alum adjuvant on the immune response was opposite in the 6–12-months-old group compared with the 1–5-years-old group, but the differences were not statistically significant.

We found that the pre-immune sera contained pre-existing *S. flexenri* 2a and *S. sonnei* IgG antibodies in all ages and the titers declined with age, as observed commonly in many other infectious diseases [29,30,31]. This observation indicated that mothers located in this endemic region probably had been exposed to *Shigella* and, in term, passively transferred *Shigella* IgG antibodies to neonates. As a result, most subjects in the study were not immunologically naive. The environmental exposure of *Shigella* in Yongfu County is probably quite diverse due to its almost quarter million population with broad social-economic status. One can anticipate a wide SD value in the pre-immune antibody level in infants. If following the theory that serum *Shigella* antibody levels may constitute a surrogate for an as yet uncharacterized mechanistic intestinal protective response, then it could also explain the epidemiological finding that the highest incidence rate of shigellosis in GuangXi is among 1- to 5-year-old children as the maternal antibodies wane [32,33]. We also noticed that there seems to be a “reversed” dosage effect in the younger groups, namely, the half-dose infants responded with slightly higher GMT than those who received the full dose. We have no satisfactory explanation except that it could possibly be due to the interference of seroconversion by maternal antibodies or immune tolerance in young infants [34].

There are several limitations of this study. The evaluation of ZF0901 immune response was based on the level of serum anti-LPS IgG elicited and we did not perform other immunological analysis, such as IgA isotype, gut-homing anti-LPS antibody-secreting cells, bactericidal activity, or other cellular immunities. The rationale of focusing on serum IgG was adopted from the well-established evidences of licensed polysaccharide vaccines (e.g., Hib, pneumococcal, and typhoid conjugate vaccines) that the critical correlation of disease protection is the level of serum IgG elicited by the vaccine. Additionally, we only compared the antibody levels between pre-immune and 30 days after the final injection. We did not keep track of the longitudinal immunological development following each inoculation, which may be of interest, especially for children under 1 years old who received multiple doses. The reason we focused on these two time points was again based on the evidence that polysaccharide antibodies approach a decent level only after the final booster shots (three times for younger and two times for older infants). We plan to collect and analyze serum samples at a later date to collect additional information on antibody persistency. Lastly, since the purpose of this study was on *Shigella* immunity, we, therefore, did not analyze TT antibody levels.

The polysaccharide vaccines are known for their non-interference character between serotypes when combined as a multivalent vaccine, as demonstrated in the recently licensed pneumococcal 20-valent PCV20 [35]. In the present study, an O-SP-based bivalent conjugate vaccine, consisting of *S. flexnera* 2a-TT and *S. sonnei*-TT, also showed to elicit robust levels of antibody to both serotypes. However, there are pros and cons of using TT as the carrier protein. TT is known to be safe in infants and has shown to be an excellent carrier. However, with the increasing number of conjugate vaccines using it as the carrier, there is a concern that children could be inadvertently overdosed with TT. To address this problem, currently there are several other potential carrier candidates, such as the newly licensed Meningococcal B outer membrane protein FHbp or the under development capsid protein VP8 from another major childhood diarrhea disease [36,37].

## 5. Conclusions

Bivalent *Shigella* conjugate vaccine ZF0901 showed to be safe and immunogenic in infants and young children. This locally manufactured vaccine candidate, following the precedent example of Vi typhoid vaccine, could potentially reduce the burden of diarrhea diseases in China and other high-endemic regions [38]. The acceptable safety profile and convincing immune response warrant an efficacy trial of the bivalent *Shigella* conjugate vaccine.

## Figures and Tables

**Figure 1 vaccines-10-00033-f001:**
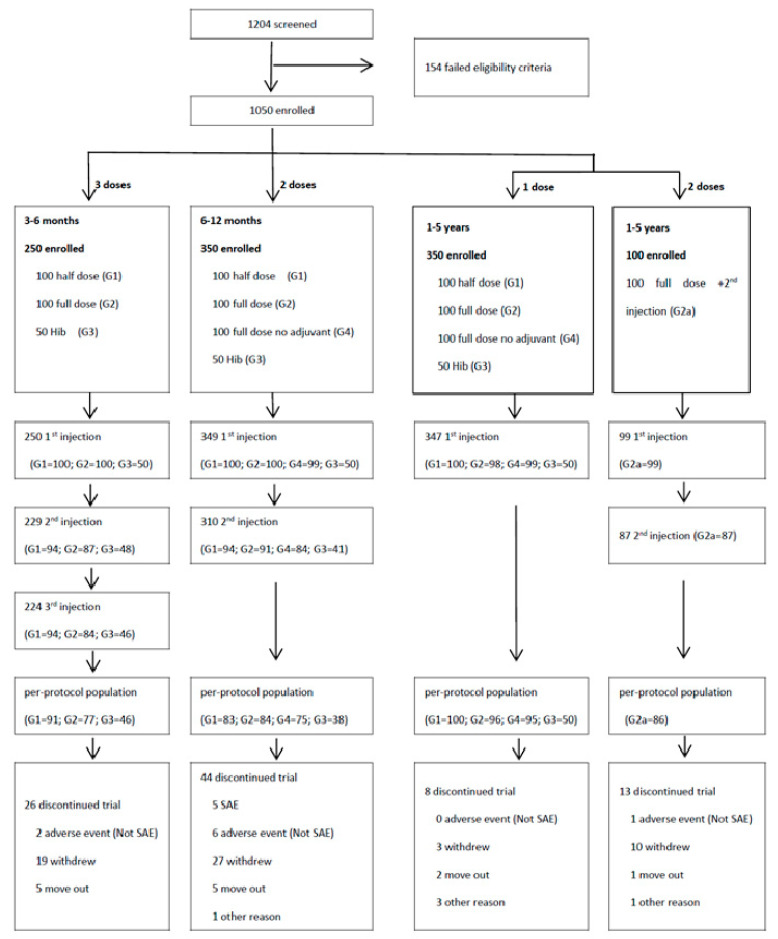
Study profile: subject randomization, screening, enrollment, and vaccine assignment of Phase II study of ZF0901 trial. SAE: serious adverse event.

**Figure 2 vaccines-10-00033-f002:**
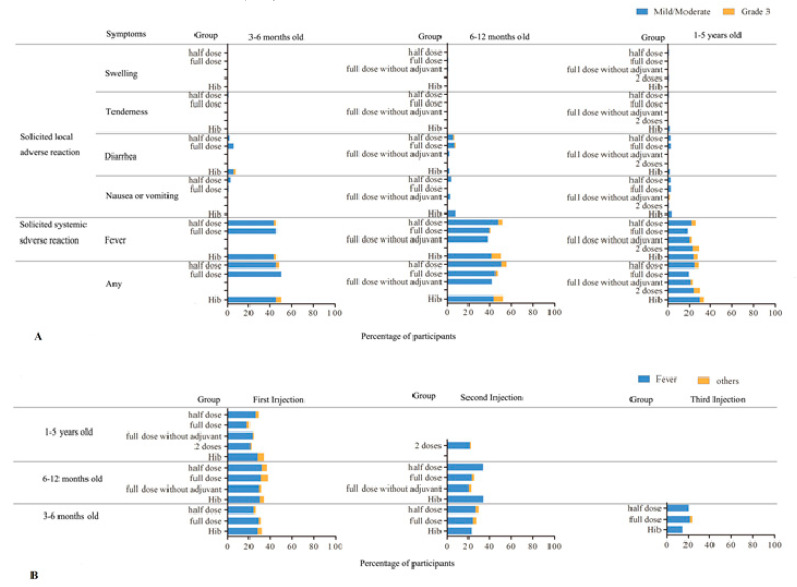
Incidence rate (%) of adverse reactions related to vaccination reported within 30 days after any injections. (**A**) Combined incidence rate according to ages. Blue bars indicate mild/moderate reactions and orange for grade 3 reactions. There was no statistical significance within the same age group receiving any type or number of vaccine injections or between any age groups (*p* > 0.5). (**B**) Comparison of the rate (%) of adverse reactions according to the number of injections and vaccine types. Blue bars indicate fever incidences and orange, for other adverse reactions. There was no statistical significance in the incidence rate between number of injections within the same age group, with or without adjuvant, full, or half dosages (*p* > 0.1).

**Figure 3 vaccines-10-00033-f003:**
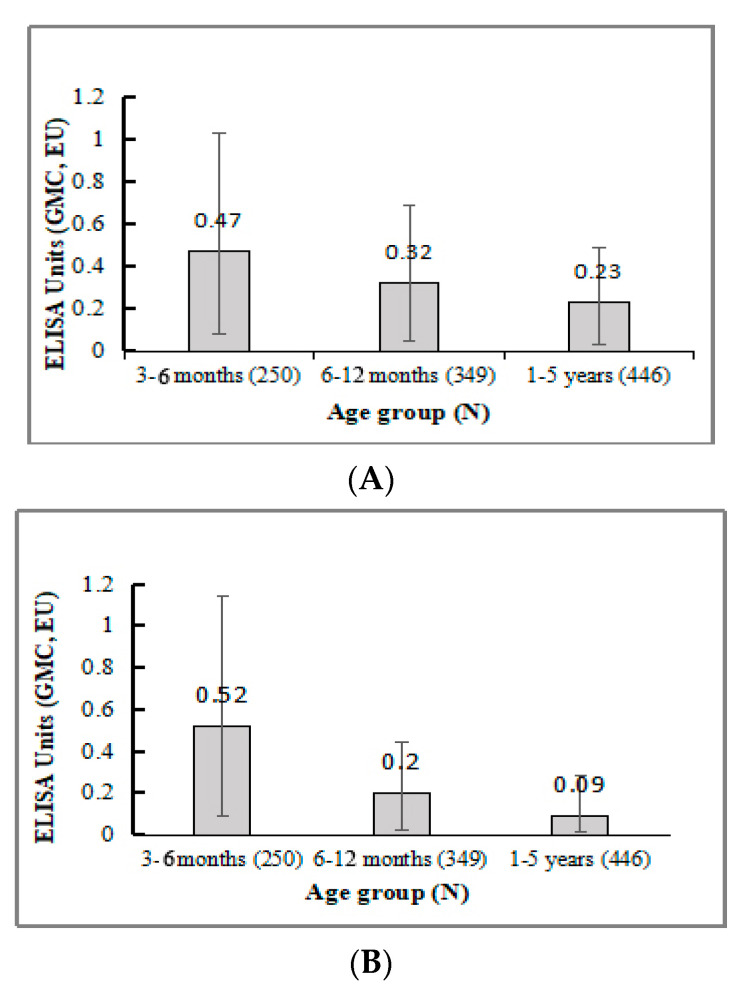
Pre-vaccination GMC anti-LPS IgG in all participants. (**A**). *S. flexneri* 2a antibody levels at day 0. (**B**) *S. sonnei* antibody levels at day 0.

**Figure 4 vaccines-10-00033-f004:**
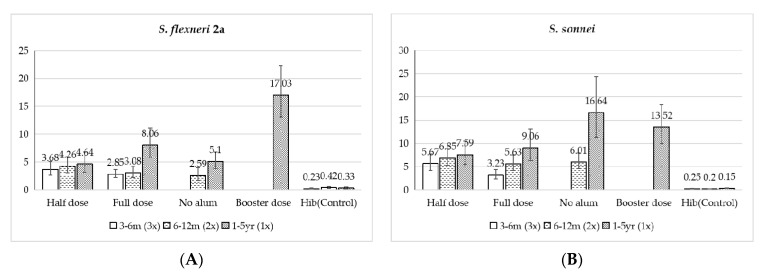
Composite chart of GMC anti-LPS IgG. (**A**) *S. flexneri* 2a, (**B**) *S. sonnei*.

**Table 1 vaccines-10-00033-t001:** Demographic Characteristics of the Participants.

3–6 Months Old	Half Dose (*n* = 100)	Full Dose (*n* = 100)	Hib (*n* = 50)		
Age at first vaccination—mo					
Mean (SD)	4.25 (0.92)	4.16 (0.91)	4.30 (0.86)		
Median	4.21	4.09	4.34		
Min, Max	3.0, 5.8	3.0, 5.9	3.1, 5.8		
Sex—no. (%)					
male	53 (53.00)	59 (59.00)	29 (58.00)		
female	47 (47.00)	41 (41.00)	21 (42.00)		
6–12 months old	half dose (*n =* 100)	full dose (*n =* 100)	full dose without adjuvant (*n =* 99)	Hib (*n =* 50)	
Age at first vaccination—mo					
Mean (SD)	8.59 (1.83)	7.71 (1.66)	8.66 (1.92)	8.61 (1.90)	
Median	8.38	7.11	8.77	8.51	
Min, Max	6.0, 11.9	6.0, 11.8	6.0, 11.9	6.0, 11.9	
Sex—no. (%)					
male	44 (44.00)	50 (50.00)	55 (55.56)	28 (56.00)	
female	56 (56.00)	50 (50.00)	44 (44.44)	22 (44.00)	
1–6 years old	half dose (*n =* 100)	full dose (*n =* 98)	full dose without adjuvant (*n =* 99)	Hib (*n =* 50)	2 doses (*n =* 99)
Age at first vaccination—yr					
Mean (SD)	2.97 (1.3)	3.01 (1.3)	3.12 (1.2)	2.83 (1.1)	2.82 (1.5)
Median	2.91	2.83	3.21	2.78	2.84
Min, Max	1.0, 5.6	1.0, 5.6	1.1, 5.9	1.1, 5.8	1.1, 5.6
Sex—no. (%)					
male	44 (44.00)	37 (37.76)	54 (54.55)	23 (46.00)	46 (46.46)
female	56 (56.00)	61 (62.24)	45 (45.45)	27 (54.00)	53 (53.54)

mo: month; yr: year; SD: standard deviation.

**Table 2 vaccines-10-00033-t002:** Serum anti-LPS IgG levels in children aged 3–6 months old vaccinated with a half or full dose of the trial vaccine ZF0901.

3–6 Months Old		Half Dose (*n =* 91)	Full Dose (*n =* 77)	Hib (*n =* 46)
*S. flexneri* 2a				
	Conversion rate (%)	58.24 (47.43, 68.50)	57.14 (45.35, 68.37)	4.35 (0.53, 14.84)
Con (EU/mL)	Pre-	0.45 (0.34, 0.59)	0.48 (0.35, 0.65)	0.52 (0.36, 0.74)
	Post-	3.68 (2.65, 5.10)	2.85 (2.19, 3.70)	0.23 (0.16, 0.33)
	Fold rise	8.26 (4.96, 13.73)	5.99 (3.67, 9.76)	0.45 (0.33, 0.62)
*S. Sonnei*				
	Conversion rate (%)	67.03 (56.39, 76.53)	53.25 (41.52, 64.71)	4.35 (0.53, 14.84)
Con (EU/mL)	Pre-	0.43 (0.33, 0.57)	0.60 (0.43, 0.83)	0.58 (0.40, 0.84)
	Post-	5.67 (4.17, 7.73)	3.23 (2.37, 4.41)	0.25 (0.18, 0.34)
	Fold rise	13.18 (7.78, 22.34)	5.38 (3.02, 9.61)	0.42 (0.31, 0.59)

**Table 3 vaccines-10-00033-t003:** Immunological response for children aged 6–12 months old with different vaccines.

6–12 Months Old		Half Dose (*n =* 83)	Full Dose (*n =* 84)	Full Dose without Adjuvant (*n =* 75)	Hib (*n =* 38)
*S. flexneri* 2a					
	Conversion rate (%)	66.27 (55.05, 76.28)	64.29 (53.08, 74.45)	64.00 (52.09, 74.77)	15.79 (6.02, 31.25)
Con (EU/mL)	Pre-	0.36 (0.27, 0.48)	0.30 (0.24, 0.39)	0.30 (0.23, 0.40)	0.29 (0.18, 0.46)
	Post-	4.26 (3.07, 5.91)	3.08 (2.22, 4.28)	2.59 (1.69, 3.97)	0.42 (0.28, 0.65)
	Fold rise	11.83 (7.99, 17.52)	10.15 (6.97, 14.77)	8.52 (5.61, 12.94)	1.48 (1.09, 2.00)
*S. Sonnei*					
	Conversion rate (%)	89.16 (80.41, 94.92)	84.52 (74.99, 91.49)	85.33 (75.27, 92.44)	10.53 (2.94, 24.80)
Con (EU/mL)	Pre-	0.19 (0.14, 0.24)	0.23 (0.17, 0.30)	0.22 (0.17, 0.30)	0.16 (0.11, 0.24)
	Post-	6.85 (5.28, 8.89)	5.63 (4.22, 7.50)	6.01 (4.57, 7.90)	0.20 (0.15, 0.27)
	Fold rise	36.47 (24.13, 55.11)	24.64 (15.96, 38.04)	27.02 (17.63, 41.42)	1.25 (0.94, 1.66)

**Table 4 vaccines-10-00033-t004:** Immunological response for children aged 1–5 years old with different vaccines and regimens.

1–5 Years Old		Half Dose (*n =* 100)	Full Dose (*n =* 96)	Full Dose without Adjuvant (*n =* 95)	Hib (*n =* 50)	2 Doses (*n =* 86)
*S. flexneri* 2a						
	Conversion rate (%)	85.00 (76.47, 91.35)	92.71 (85.55, 97.02)	85.26 (76.51, 91.70)	4.00 (0.49, 13.71)	95.35 (88.52, 98.72)
Con (EU/mL)	Pre-	0.18 (0.13, 0.25)	0.23 (0.17, 0.32)	0.24 (0.17, 0.33)	0.28 (0.18, 0.42)	0.25 (0.18, 0.34)
	Post-	4.64 (3.17, 6.82)	8.06 (5.84, 11.13)	5.10 (3.81, 6.83)	0.33 (0.20, 0.54)	17.03 (13.04, 22.24)
	Fold rise	25.75 (18.11, 36.63)	35.04 (25.83, 47.53)	21.50 (15.58, 29.67)	1.20 (0.88, 1.64)	68.15 (47.87, 97.02)
*S. Sonnei*						
	Conversion rate (%) PPS	95.00 (88.72, 98.36)	94.79 (88.26, 98.29)	95.79 (89.57, 98.84)	6.00 (1.25, 16.55)	96.51 (90.14, 99.27)
Con (EU/mL)	Pre-	0.09 (0.07, 0.11)	0.08 (0.06, 0.10)	0.11 (0.08, 0.14)	0.09 (0.07, 0.13)	0.09 (0.07, 0.12)
	Post-	7.59 (5.49, 10.50)	9.06 (6.28, 13.09)	16.64 (11.34, 24.42)	0.15 (0.10, 0.22)	13.52 (10.01, 18.26)
	Fold rise	85.99 (59.79, 123.68)	112.06 (76.16, 164.87)	158.03 (102.69, 243.22)	1.55 (1.22, 1.96)	146.01 (103.11, 206.74)

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
