# Peer review of "Safety and Immunogenicity of a Shigella Bivalent Conjugate Vaccine (ZF0901) in 3-Month- to 5-Year-Old Children in China"

_vaccines, 2021, doi:10.3390/vaccines10010033_

Round 1

Reviewer 1 Report

The manuscript by Du and coworkers details a phase II study of a conjugate vaccine (ZF0901). This vaccine was manufactured by the team and screened in hundreds of children in China. Thus far, only minor adverse effects were funded.

The work is solid and merits publication. However, define (SAE) in figure 1 and add a paragraph before the conclusion on what are the steps and requirements to go to phase III clinical trials as stated by the CDC or the agency in charge. 

Author Response

1. Define SAE as Serious Adverse Event below the chart of figure 1.

2. Added a paragraph before the conclusion on what are the steps and requirements to go to phase III clinical trials as stated by the CDC or the agency in charge.

Clinical phase II suspension/termination criteria: there are serious adverse events related to the vaccine that lead to the death or life-threatening of the subject; the adverse reactions of grade 3 and above related to vaccination in each dose of each age group determined by the researcher exceed 15%. According to the clinical protocol, without the above suspension/termination criteria, it can enter phase III clinical trial.

We re-describe the conclusion to see if we can express a similar meaning. Please see manuscript.

Please see attached revised manuscript.

Reviewer 2 Report

The manuscript of Mo et al. study the safety and immunogenicity of a Shigella Bivalent conjugate vaccine (O-specific polysaccharides of S. flexneri and S. sonnei covalently bound to tetanus toxoid) in 3 months to 5 years old Children in China. The effect of adjuvant and dosage (full or half dose) was investigated. The authors demonstrate that the vaccine is safe and immunogenic, being suitable for routine immunization.

Major comments:

  • It is not clear why each group of age have a different protocol of vaccination. Example: Cohort 1 has 3 injections, cohort 2 has 2 injections, the majority of corhort 3 have 1 injection and a smaller group have 2 injections. Please include an explanation for these different protocols for each group to be more clear.

Minor comments:

  • Introduction, line 54-56: Please rephrase this sentence, to be clear that the flexneri 2a conjugate is able to induce the type 2a and cross-reactive type 6 antibodies
  • Introduction, line 59-61: The information giving on this phrase is similar to the previous information giving on lines 48-49. Please review these 2 phrases.
  • Results, line 226: “…959 subjects completed the trial.” However, in the abstract is a total of 968 children (224, 310 and 434) described. Please review if these numbers are correct.
  • Figure 1: To be more clear the different groups and conditions tested, the cohort 3 (1-5 years) should be divided in two in this figure representation, the group with 1 injection and the group with 2 injections (0 and 30 days).
  • Table 1: Please correct “famale” to “”female”.
  • Figure 2: this figure is too small, difficult to interpret. Please improve the quality of these 2 images
  • Figure 3: The SD is too high, meaning that the antibody titers are too much diverse in the same group age. Please take this in consideration on the analysis of these pre-vaccination data.
  • Reference – REF 10 is not mentioned on the text.

Author Response

1. It is not clear why each group of age have a different protocol of vaccination. Example: Cohort 1 has 3 injections, cohort 2 has 2 injections, the majority of corhort 3 have 1 injection and a smaller group have 2 injections. Please include an explanation for these different protocols for each group to be more clear.

No shigella conjugate vaccine trials were conducted in infants younger than 1 year old so far. We believe that conjugate vaccines might have similar immunological characteristics, so we designed the protocol according to the immunization procedure of other marketing conjugate vaccines (Hib, Pnum and Men) in China.

Basic immunization program of conjugate vaccine in China

3-6 months old

6-12 months old

1-2 years old

2-5 years old

Hib

3 injections, 1 mo apart

2 injections, 1 mo apart

1 injections

1 injections

Pnum conjugate

3 injections, 1 mo apart

2 injections, 2 mo apart

2 injections, 2 mo apart

1 injections

Men conjugate (A+C)

3 injections, 1 mo apart

2 or 3 injections, 1 mo apart

2 injections, 1 mo apart

1 injections

    There were shigella conjugate vaccine trials conducted in children over 1 year old, the immunization regime were 2 injections with 6-week intervals, some results showed boost reaction for 2 dose, some did not (references as below). So we got a small group of children over 1 year old having 2 injections to see the difference with 1 injection.

S Ashkenazi, et al. Safety and Immunogenicity of Shigella sonnei and Shigella flexneri 2a O-Specific Polysaccharide Conjugates in Children. J Infect Dis. 1999 Jun;179(6):1565-8. doi: 10.1086/314759.

Justen H. Passwell, et al. Age-related efficacy of Shigella O-specific-polysaccharide conjugates in 1 to 4 year-old Israeli children. 2010 Mar 2;28(10):2231-2235. doi: 10.1016/j.vaccine.2009.12.050.

Justen H Passwell, et al. Safety and immunogenicity of Shigella sonnei-CRM9 and Shigella flexneri type 2a-rEPAsucc conjugate vaccines in one- to four-year-old children. Pediatr Infect Dis J. 2003 Aug;22(8):701-6. doi: 10.1097/01.inf.0000078156.03697.a5.

2. Introduction, line 54-56: Please rephrase this sentence, to be clear that the flexneri 2a conjugate is able to induce the type 2a and cross-reactive type 6 antibodies

Rephrase as required, please see manuscript.

3. Introduction, line 59-61: The information giving on this phrase is similar to the previous information giving on lines 48-49. Please review these 2 phrases.

Line 59-61 were deleted, please see manuscript.

4. Results, line 226: “…959 subjects completed the trial.” However, in the abstract is a total of 968 children (224, 310 and 434) described. Please review if these numbers are correct.

These numbers are correct. 968 children (224, 310 and 434) finished vaccination, a few of them withdraw from blood taken or safety monitor, only 959 completed the trial.

5. Figure 1: To be more clear the different groups and conditions tested, the cohort 3 (1-5 years) should be divided in two in this figure representation, the group with 1 injection and the group with 2 injections (0 and 30 days).

The figure was modified, please see manuscript.

6. Table 1: Please correct “famale” to “”female”.

The error was corrected, please see manuscript.

7. Figure 2: this figure is too small, difficult to interpret. Please improve the quality of these 2 images

The figure was improved, please see manuscript.

8. Figure 3: The SD is too high, meaning that the antibody titers are too much diverse in the same group age. Please take this in consideration on the analysis of these pre-vaccination data.

Added in discussion, please see manuscript.

9. Reference – REF 10 is not mentioned on the text.

Being added, please see manuscript.

Reviewer 3 Report

The manuscript of Mo et al. has performed a Phase II, double blinded vaccine clinical against Shigella among 3 months to 5 years old children in China. The vaccine has LPS from 2 popular species covalently bonding to tetanus toxoid. The trial recruited more than 900 children who were enrolled into 3 different age groups. The results showed a good immunogenicity elicited by the vaccine, as well as a low side effect. In general, the paper is well written. However, several concerns should be addressed before it is accepted as the current form.

  1. Line 420 to 435 could be moved to the background, since the benefits of using O-antigens and TT should be mentioned first. It will justify the reason to use such methods, and send the message to the readers.
  2. The reason to choose YongFu county as the place to perform the trial could be discussed, such as high prevalence of disease?
  3. GMC was not defined in the paper (line 27). It indicates geometric mean concentration, but was not mentioned in the text.
  4. Line 208, the name of the statistical method should be “Clopper-Pearson”.
  5. Line 215, the author may want to explain the definition of “MedDRA” to the audience, as well as the reference.
  6. Line 439, the author concluded the vaccine is suitable to be included in routine immunization, which is too strong. The trial is still in Phase II, even the immunogenicity is remarkably increased, it is too early to tell the protection efficiency.
  7. Some sentences in the manuscript is tedious. Such as line 53, it could be expressed as “In Africa and Asia”.

Author Response

1. Line 420 to 435 could be moved to the background, since the benefits of using O-antigens and TT should be mentioned first. It will justify the reason to use such methods, and send the message to the readers.

Line 420 to 435 were moved to the background, please see manuscript.

2. The reason to choose YongFu county as the place to perform the trial could be discussed, such as high prevalence of disease?

We did not choose Yongfu as the site according to the epidemic situation of the disease, but according to the work level of the site personnel. Yongfu belong to Guangxi, which is the most experienced region in vaccines trial in China.

3. GMC was not defined in the paper (line 27). It indicates geometric mean concentration, but was not mentioned in the text.

GMC was defined as comments, please see manuscript.

4. Line 208, the name of the statistical method should be “Clopper-Pearson”.

The error was corrected, please see manuscript.

5. Line 215, the author may want to explain the definition of “MedDRA” to the audience, as well as the reference.

Explained the definition of “MedDRA”, please see manuscript.

6. Line 439, the author concluded the vaccine is suitable to be included in routine immunization, which is too strong. The trial is still in Phase II, even the immunogenicity is remarkably increased, it is too early to tell the protection efficiency.

The sentence has been deleted, please see manuscript.

7. Some sentences in the manuscript is tedious. Such as line 53, it could be expressed as “In Africa and Asia”.

Rephrase as required, please see manuscript.

Please check attached revised manuscript.

Reviewer 4 Report

Mo et al. analysed the safety and immunogenicity of a Shigella bivalent conjugate vaccine in 3 months to 5 years old children in China. The candidate vaccine is based on O-specific polysaccharides purified and detoxified from lipopolysaccharide of S. flexneri 2a and S. sonnei and covalently bound to tetanus toxoid. They divided their cohort in three sub-cohorts: 3-6 months, 6-12 months and 1-5 years old children. According to the sub-cohort, children received between one and three injections. The authors evaluated the safety and immunogenicity of half, full dose, full dose without adjuvant or Hib vaccine as control. They analysed the adverse reactions and S. flexneri 2a and S. sonnei LPS-specific IgG antibody responses in serum on days 0 and 30 days following the last injection by ELISA to assess the immunogenicity of the candidate vaccine. The candidate vaccine looks safe in all groups and is immunogenic especially in 1-5 years old children but age-dependent IgG responses were observed for both Shigella serotypes.

The development and assessment of Shigella candidate vaccines are crucial as so far there is no licensed vaccine available. This Phase II clinical trial of a Shigella bivalent conjugate vaccine is very relevant. I think the multivalence is an asset for this type of candidate vaccine. The evaluation of safety and immunogenicity of this candidate vaccine looks good to me. The design of the study, the endpoints as well as the ELISA method look sensible. The only point I would like to discuss is the lack of data showing the functionality of antibodies generated by the candidate vaccine. Of course, it is a little bit disappointing not to have any data about intestinal immune responses. For example, IgA responses in saliva which could give an indication about intestinal IgA responses.

Major comments:

Line 278: How did the authors define the cut-off (LLOD) of the ELISA ?

Could the authors analyse the functionality of IgG antibodies generated post-vaccination ? For example, serum bactericidal activity. I think it would strengthen the message to show that the vaccine-induced antibodies are functional. 

This study does not give any information about mucosal responses induced post-vaccination, it is a limitation of this study which is well explained in the discussion. But could the authors analyse IgA responses in serum ? Sometimes it does not reflect IgA responses in mucosal surfaces but it might give an idea.

Minor comments:

Line 21: “bound to” ?

Line 49: “including several using”, problem of sentence ?

Line 53: the word should probably be “predominant” ?

Line 118: “one” instead of “on” ?

Line 127: the abbreviation CRF to be defined

Line 141: Is there any reference to show that the residual endotoxicity in the final product was lower than the safety requirement set by NMPA ?

Line 156: For the sub-cohort 2, it is written in the text that they are 6-12 months old children and in the Figure 1, it is written 6-11 months old children. Can the authors clarify ?

Line 178: red dot to be modified.

Figure 1: the quality of this figure is not very good. Could the authors improve it ?

Do the authors know if there is a cross-reactivity between S. flexneri 2a and S. sonnei IgG antibodies ?

Author Response

1. Line 278: How did the authors define the cut-off (LLOD) of the ELISA ?

The specific concentrations of IgG antibody were measured by ELISA with reference sera as standard, reference sera were pooled from adult donors and assigned to contain 100 ELISA units (EU) anti-LPS IgG per milliliter. The test sera and the reference sera were diluted in parallel for 8 wells. After reading, the relative concentration of the test sera was calculated by linear regression analysis.

2. Could the authors analyse the functionality of IgG antibodies generated post-vaccination ? For example, serum bactericidal activity. I think it would strengthen the message to show that the vaccine-induced antibodies are functional.

The clinical protocol only designed ELISA antibody detection. We totally agree with your suggestion and will do functional antibodies detection in the follow-up clinic to enrich the clinical data.

3. This study does not give any information about mucosal responses induced post-vaccination, it is a limitation of this study which is well explained in the discussion. But could the authors analyse IgA responses in serum ? Sometimes it does not reflect IgA responses in mucosal surfaces but it might give an idea.

The clinical protocol only designed ELISA antibody detection, we agree with your suggestion. Same with comments 2, we could not do extra detection because the protocol was approved by the ethics committee and filed with the national genetic office, it could not be changed after finishing the trial. We will increase the test content according to your suggestions in the follow-up clinical research.

4. Line 21: “bound to” ?

This is usually described in conjugate vaccines

5. Line 49: “including several using”, problem of sentence ?

The error was corrected, please see manuscript.

6. Line 53: the word should probably be “predominant” ?

The error was corrected, please see manuscript.

7. Line 118: “one” instead of “on” ?

The error was corrected, please see manuscript.

8. Line 127: the abbreviation CRF to be defined

CRF was defined, please see manuscript.

9. Line 141: Is there any reference to show that the residual endotoxicity in the final product was lower than the safety requirement set by NMPA ?

Rephrase the sentence, please see manuscript.

10. Line 156: For the sub-cohort 2, it is written in the text that they are 6-12 months old children and in the Figure 1, it is written 6-11 months old children. Can the authors clarify ?

The error was corrected, please see manuscript.

11. Line 178: red dot to be modified.

Be modified, please see manuscript.

12. Figure 1: the quality of this figure is not very good. Could the authors improve it ?

Be improved, please see manuscript.

13. Do the authors know if there is a cross-reactivity between S. flexneri 2a and S. sonnei IgG antibodies ?

The structure of S. flexneri 2a and S. sonnei O-specific polysaccharide distinguished with each other, no cross-reactivity between S. flexneri 2a and S. sonnei IgG antibodies.

Please check attached revised manuscript and the structures of the polysaccharide

Sorry, only 1 ducument could be attached. If you want the structure, I'll get later.

This manuscript is a resubmission of an earlier submission. The following is a list of the peer review reports and author responses from that submission.